# Vibration-Based Smart Sensor for High-Flow Dust Measurement

**DOI:** 10.3390/s23115019

**Published:** 2023-05-24

**Authors:** Anibal Reñones, Cristina Vega, Mario de la Rosa

**Affiliations:** CARTIF Technology Center, Av. Francisco Vallés, 4, 47151 Boecillo, Spain; aniren@cartif.es (A.R.); marros@cartif.es (M.d.l.R.)

**Keywords:** sensors, innovation, process industry, automation, industry 4.0, digital transformation, industrial plants, filler, dust, vibration, signal processing, smart sensing

## Abstract

Asphalt mixes comprise aggregates, additives and bitumen. The aggregates are of varying sizes, and the finest category, referred to as sands, encompasses the so-called filler particles present in the mixture, which are smaller than 0.063 mm. As part of the H2020 CAPRI project, the authors present a prototype for measuring filler flow, through vibration analysis. The vibrations are generated by the filler particles crashing to a slim steel bar capable of withstanding the challenging conditions of temperature and pressure within the aspiration pipe of an industrial baghouse. This paper presents a prototype developed to address the need for quantifying the amount of filler in cold aggregates, considering the unavailability of commercially viable sensors suitable for the conditions encountered during asphalt mix production. In laboratory settings, the prototype simulates the aspiration process of a baghouse in an asphalt plant, accurately reproducing particle concentration and mass flow conditions. The experiments performed demonstrate that an accelerometer positioned outside the pipe can replicate the filler flow within the pipe, even when the filler aspiration conditions differ. The obtained results enable extrapolation from the laboratory model to a real-world baghouse model, making it applicable to various aspiration processes, particularly those involving baghouses. Moreover, this paper provides open access to all the data and results used, as part of our commitment to the CAPRI project, with the principles of open science.

## 1. Introduction

The development of the smart sensor presented in this paper is part of the CAPRI H2020 project [1,2], wherein one of the main objectives is to develop cognitive solutions for the process industry and a cognitive automation platform (CAP), with the final objective of contributing to the digital transformation of process industries. The CAPRI project enables cognitive tools to provide the existing process industries with flexibility of the process, improving the quality control of products and intermediate flows.

One of the main challenges for a process industry is to enable efficient monitoring and control when the production process or environments are complex, e.g., due to harsh temperature conditions that a system is operating in, or the abrasive nature of the environment in a process. An example of these harsh conditions is the required measurement of a high concentration of dust flow through a pipe, as in the case of a baghouse used in asphalt mix manufacturing. In this paper, the authors propose an innovative smart sensor to measure the flow of dust that can be used with a high concentration of particles, based on the analysis of the vibrations produced along a slim bar inside the pipe of the baghouse. The authors of the paper found some commercial sensors, during the first stages of the development of the CAPRI project [3,4,5,6], able to measure flows inside pipes, but none were designed for the harsh conditions inside a baghouse, such as high temperature (230 °C), high concentration (200 g/m^3^), high velocity of the particles (45 m/s) and the abrasiveness of the particles that can be found, for example, in the manufacturing of asphalt mixes.

The asphalt plant involved in the CAPRI project has four main manufacturing steps: (1) it starts in the cold bins, where the asphalt process starts dossing the raw materials classified in five bins by size. (2) The next processing step is the drier drum (the key process in this paper), which extracts all the humidity present in the cold aggregates, and the finest dust particles—which are the smallest, being less than 63 μm—called filler, through the baghouse aspiration pipe. All the materials that make up asphalt contain filler in different proportions, while the finest portion of the asphalt mix, the sand, contains the highest percentage, around 90% of the weight of the filler. Meanwhile, in the percentage on the whole cold aggregates is around 10% by weight. The dried material is stored (3) in the hot bins classified by size and then mixed (4) with the rest of asphalt mix components.

These components, by weight, consist of 95% of gravel, sand and filler. The other 5% comprises agents to bind the aggregates, such as the bitumen. In Figure 1a scheme of the drying process, including the actual sensor infrastructure used for monitoring and controlling the drying process, is presented with brown arrows for the flows of aggregates, and blue arrows for the flows of air. The cold aggregates enter the direct-fired rotary dryer-drum where they are heated and dried, with the majority of the filler dust being aspirated with the flue gases. These gases are transported towards the baghouse and the filler is retained within the set of the fabric bags and the clean flue gas exits through the chimney to the atmosphere, without the dust particles. Since the hot-mix asphalt recipe specifies the content of dust filler in the final mixture, which is larger than the amount of dust retained in the dried aggregates, additional filler has to be injected in the mixing tower. This injected filler will be either commercial dust or part of the extracted dust recovered in the bag filter.

It can be observed that the manufacturing process involves heating the filler, extracting it and, in the final step, once again adding new cold filler in the final mix, which causes wasting of energy in the rotary dryer drum and in the filtering (baghouse) process. The process of heating the filler and then extracting it has an associated loss of energy, because the finest particles requires more energy to eliminate their humidity than the largest particles, as presented in the work of Ang et al. [7] on the analysis of process energy use in asphalt-mixing plants. With a sensor that would measure the mass flow of filler extracted in the baghouse in real time, the drying process would extract only the excess of filler not needed in the final asphalt mix with a saving of thermal energy.

Thus, the objective of the sensor presented in this paper is to measure the flow of filler, or the finest dust present in the cold aggregates during the drying process. Once the authors analyzed the working conditions of the asphalt manufacturing process and the unavailability of commercial sensors appropriate to such conditions, they undertook a search for the most advanced approaches to similar or related monitoring problems. The basic criteria was scientific works related to vibration analysis in similar aspiration processes and approaches aiming to measure granulometry or mass flow of particles inside pipes. Based on these search criteria, the authors found examples in various papers, such as techniques applied using neural networks with image processing, to estimate the granulometric distribution of small- and medium-sized aggregates [8]; the use of machine vision to analyze particulate material and conveyor belts [9]; new techniques based on tracking velocimetry in pipe flows [10]; miniaturized sensors based on nanofibers that determine vibrations and analyze possible flow in different structures [11]; laser technology using a time-of-transition technique for the measurement of size distribution [12]; and new capacitive sensors using a tomographic calibration-based approach to measure particle flow in pipes [13]. After analyzing all these works, the authors found interesting approaches; however, none were developed under the harsh conditions present in the asphalt mixing process, such as abrasive dust, high exhaust gas temperature (230 °C), high exhaust gas velocity (45 m/s) and high dust concentration (200 g/m^3^). Another technology for mass flow measurement, found in the literature, was based on vibration analysis of a gauge plate, and FFT through an LSTM neural network [14]. The solution shown in that work would imply that the installation of a barrier, across the total flow in the aspiration pipe of the baghouse, would affect the whole aspiration process. A non-intrusive or low-intrusion measurement process is needed to estimate the filler flow in the asphalt manufacturing process analyzed by the authors.

In addition, to look for solutions within scientific works, the authors analyzed the patent landscape for similar solutions. Inspired by a similar approach as the one used with so-called impact flow meters used for granular materials—which use an impact force produced by falling material for flow meter measurement (as found in different patents [15,16])—the authors decided to measure the generated vibration along a pipe during filler aspiration. After the initial laboratory experimentation, it was determined that there were vibration events related to different filler aspiration moments that could be correlated to the flow mass that was collected at a given time. Figure 2 depicts the vibration registered during such experimentation using the laboratory setup presented in Section 2. The figure represents the real-time vibration in the Figure 2a and the time-frequency analysis (spectrogram) of such vibration in the Figure 2b.

At the interval from 0 to 6 s, the vibration levels are related to aspiration equipment operation only. After 6 s, the values of vibration above 4 kHz are registered as wide-band noise vibration, which corresponds to the filler aspiration events. These events are then registered in three bursts that correspond to an aspiration of three different quantities of filler. Below that frequency range, the mechanical vibrations caused by the aspiration equipment hide any vibration event related to the aspiration of filler. It is very important to perform an exhaustive frequency analysis, to detect which are the frequency components associated with the aspiration of filler.

Once the vibration is acquired, fast Fourier transform allows the conversion of the signal from time domain to frequency domain, and analyzes the areas of the spectrum where vibration due to the filler aspiration occurs. Once the FFT is calculated, the energy value (in g_RMS_) between an initial and final frequency can be calculated. The main idea is to relate this energy value with the aspirated filler flow. Figure 3 depicts the vibration level, or RMS, in four selected frequency bands: 0–3.4 kHz (white), 6–8 kHz (Red), 9–13 kHz (green) and 14–20 kHz (blue). The experimentation shown allowed the authors to select the band from 14 to 20 kHz (blue in Figure 3) as the band with the highest signal to noise ratio between the vibration of the equipment and the vibration of the filler aspiration events. The proposal the authors introduce is that the vibration energy in that frequency band should be an estimation of filler mass flow in real time. In the next sections, the method used and the experimental proof of concept, undertaken to demonstrate the feasibility of such measurement, are described. 

## 2. Materials and Methods

During the development of this sensor, various steps were taken. In the early stages, a vibration sensor was installed at different points of a laboratory prototype, to determine whether there were one or more locations sensitive to events related to the aspiration of filler. The tests were performed by installing an accelerometer in a slim steel bar across the pipe. The aim in using the bar was to evaluate whether a similar setup could be feasible using the corresponding PT100 sensor bar of an actual baghouse pipe present in the asphalt plant.

The measurements and tests were performed by designing a scale prototype, to simulate the conditions of concentration and velocity of the flow inside the pipe of a baghouse of an asphalt plant. The laboratory prototype is shown in Figure 4.

This setup has eight main parts, all of which are commercially available but have never been used for this kind of experiment. The first is an industrial dust vacuum (1) to create the aspiration depression, which is connected to a pipe (3), where the accelerometer (2) is installed (a commercial sensor from PCB, model 352C65, with a sensitivity of 100 mV/g) shown in Figure 5. The pipe (3) has a hole in which to insert a slim steel bar (4) (35 mm × Ø 3.75 mm). The purpose of the bar is to behave similarly to the PT100 installed in the actual asphalt plant baghouse, where another accelerometer would be installed. This bar helps to fix the accelerometer and measure the collisions and vibration caused by the flow of filler through the pipe. The dust is transported to the pipe by a conveyor belt (5), with the amount and distribution of filler (6) needed depending on the experiment to be performed. The final part of the laboratory prototype is the data acquisition system (7) from National instruments [17], a 24-bit vibration input module (NI 9234) connected to a laptop (8), to measure the vibration sensed by the accelerometer.

The first step is to calibrate the sensors, measuring the vibration without filler, to obtain the vibration base value. This value has to be used to compensate for the vibration generated by the vacuum equipment, including its electrical motor, plastic hose and the installed pipe, when the aspiration takes place without the flow of filler through the pipe. In the actual plant, this occurs at the beginning and the end of each drying cycle; in this paper, it is represented in a laboratory setting, aspirating without filler. The vibration analysis has been performed, analyzing the main frequency bands. That is, 0–3.4 kHz, 6–8 kHz, 9–13 kHz and 14–20 kHz. The last band, 14–20 kHz, contains the highest signal to noise ratio of the sensor, with the aspiration with or without filler. Figure 6 shows the vibration RMS value in the frequency band of 14–20 kHz, which is constant during the whole vacuum data acquisition, and which must be eliminated from the rest of the experiments in order to obtain the vibration energy due only to the filler aspiration. 

To obtain the estimated filler flow, the procedure is based on the integration of the vibration energy value (in g_RMS_) in the selected frequency band (14–20 kHz), and a comparison with the known total mass aspirated from each experiment. The ratio of these values corresponds to the conversion constant *C* that must be used to convert the vibration energy from “g_RMS_” to “grams” of filler per second (see Equations (1) and (2)).
(1)$$Flow\;mass=\frac{C\times Energy}{time}$$
(2)$$C=\frac{Total\;mass\;from\;experiments}{Total\;energy\;from\;14,000–20,000\;\mathrm{Hz}\;band\;from\;experiments}$$

After converting the vibration energy to mass flow, it is necessary to integrate it along the time of aspiration, to obtain the real amount of filler aspirated during the experiment.

The methodology used for the filler aspiration experiments consists of creating a vacuuming, followed by a complete clean of its filter, to prevent filler accumulation on the filter from affecting the measured vibration and altering the results obtained.

The experiments were performed with the help of a mass scale, to measure exactly the amount of filler mass to be aspirated. It is worth noting that all the experiments were performed in a laboratory, using the same kind of dust filler used in an actual asphalt plant. Figure 7, shows the mass balance used during the experiments.

The design of experiments for validation is always a difficult point in each analysis of prototypes, and the authors finally selected this series of experiments to be more representative of various distributions of filler:Aspiration of a continuous strand of filler (Figure 8a); The filler is distributed along the conveyor, with the same amount of filler per unit length, to aspirate homogeneously the filler.Aspiration of small consecutive piles of filler (Figure 8b); the filler is distributed along the conveyor in three equal piles, to demonstrate that the sensor is sensitive to the aspiration of filler with start & stop amount of filler.Aspiration of an increasing amount of filler (Figure 8c); the filler is distributed from less to more amount along the conveyor, to relate the increase of vibration with the increase of filler.Aspiration of a decreasing amount of filler; the filler is distributed from more to less amount along the conveyor, to relate the decrease of vibration with the decrease of filler.

In the next section, the results of the different experiments and the comparison of the estimated mass of filler are presented, with the procedure discussed as well.

## 3. Results

To obtain good results and ensure their replicability, several experiments have been conducted, varying the amount of filler and the distribution of mass along the aspiration process, with different distributions of particles and with the same weight, to guaranty the stability of the measurement process. Each experiment’s result is presented using three graphs, the first one being a time–frequency graph, or spectrogram, of the vibration acquisition of the experiment, the second representing the vibration energy of the four energy bands analyzed (0–3.4 kHz white, 6–8 kHz in red, 9–13 kHz in green and 14–20 kHz in blue) and lastly, a simulation of the accumulated weight of filler aspirated—based on the procedure of processing the vibration energy in the 14–20 kHz band and transforming it into a mass flow (g/s). This last graph also contains the final total estimated mass of each experiment.

### 3.1. First Experiment

The filler in this first experiment was distributed uniformly along 10 cm of conveyor, with a total weight of 25 g. In Figure 9 and Figure 10, the aspiration starts in the first second, and finishes in the fourth second of analysis. With only 3 s of aspiration, a spike can be seen in the frequency band between 14–20 kHz, caused by the onset of filler aspiration. The frequency band of 0–3.4 kHz does not clearly show an aspiration, since the vibration of the motor masks the vibration related to the aspiration of the filler. Figure 11 depicts the accumulated mass of filler aspirated, and the final estimated value of 24.63 g, this measure having an error of 1.48% in comparison to the 25 g, disposed in the conveyor, to be aspirated. This simulation has been carried out using Equations (1) and (2), and integrating the mass flow obtained with them.

### 3.2. Second Simulation Experiment

In the second experiment, the filler is distributed in 4 piles, with very similar but not identical amounts of dust, with a total mass of 60 g. The spectrogram in Figure 12 clearly shows the vibration events during aspiration of each pile, which are also visible in the frequency bands shown in Figure 13. In Figure 14 it can be seen that the total estimated mass is 58.92 g, with an error of 1.80% over the actual quantity aspirated.

### 3.3. Third Simulation Experiment

Figure 15 and Figure 16 show the time–frequency vibration analysis of the third experiment, with 100 g of filler increasingly distributed along 40 cm. Figure 17 shows the estimation of accumulated filler and the total final estimated quantity of 101.85 g, with an error of 1.85%, in comparison with the actual weight measured on the scale.

The increase in mass, collected over time, is correlated with the increase in vibration measured by the sensor, starting at second 1 and continuing until second 12.

### 3.4. Fourth Simulation Experiment

The last experiment is performed with the aspiration of 100 g of filler, decreasingly distributed along 40 cm. Figure 18 and Figure 19 show the vibration analysis and Figure 20 represents the simulation of accumulated mass aspirated, and the final value of total filler mass (102.56 g), which equals a 2.56% error, in comparison with the 100 g, disposed in the conveyor, to be aspirated.

The simulation demonstrates the increase in flow from second 2, where the aspiration starts, until second 11, where it ends. The aspirated weight increases in a more pronounced way at the beginning and decreases over time, with the filler distribution decreasing as well.

### 3.5. Experiments’ Results, Discussion and Replicability

In the four experiments, it is demonstrated that the value obtained in the simulation of the aspirated mass of filler is very close to the weighed value. This proves the success of the dust flow estimation by measuring the vibration caused by the filler flow through a pipe with an accelerometer on a slim bar. Table 1 shows the comparison, for each experiment, of actual and estimated weight, showing an error of less than 3%.

All the experiments were repeated several times, to determine whether the results obtained were the same with different amounts of filler (10, 20 and 30 g). With these experiments, the reproducibility of the results can be seen, as presented in this paper. Figure 21 represents these experiments with various amounts of filler in the aspiration of a continuous strand of filler (Figure 8a). The graph contains the relation among the vibration energy (in g_RMS_) and the total aspirated mass in the selected frequency band (14–20 kHz):

The graph demonstrates that the results, of measuring the amount of filler aspirated, do not depend on the amount and distribution of filler. The errors of the measurement are less than 6%, which is higher than the 3% of error shown in Table 1. In the case of this repeatability test, the cleaning of the filter was only performed in one out of two instances, which caused the increased error.

## 4. Conclusions

The laboratory sensor prototype shown in this paper enables the real-time measurement of the amount of aspirated filler flow through a pipe, with an error of less than 6%. The advantage of the proof of concept shown is that it can be used in harsh conditions, such as high temperature, abrasive particle flow, high pressure and high concentrations of particles. This is because the materials used can resist such conditions, and, in case of damage, the test probe (a slim steel bar) can be replaced easily, without additional calibrations. The experiments conducted, with various conditions in the distribution of filler (distributed, accumulated in piles, etc.), ensured the replicability of the results and the stability of the overall measurement and filler flow estimation procedures, with the latter being contingent on the cleanliness of the filter. In the case of an industrial plant, the cleaning is conducted automatically, whereas in laboratory conditions, this must be performed manually.

The first step in using this measurement principle is to choose the bands where a difference in vibration energy is more apparent and related to filler aspiration events. The calibration is easy to do at laboratory conditions, knowing the aspirated filler mass to create the model for estimating the mass flow passing through the pipe. It is essential to be able to only detect the frequency bands, in which this variation is noticeable. Knowing the mass aspirated in a test for the calibration of the system, an approximate model can be made to transform the vibration measured into the filler aspirated by the industrial vacuum cleaner.

As the sensor is based on the analysis of vibrations produced by the flow of particles crashing into a slim bar, it could also be applied to other aspiration processes, mainly within baghouses. Different baghouse installations will have different vibration patterns that will be reflected in different areas of the vibration spectrum. This necessitates a calibration of each system, to select the appropriate frequency band with the highest signal to noise ratio between the vibration of the vacuum equipment and the vibration due to the filler aspiration events.

The results obtained allow an extrapolation between the model obtained in the laboratory and a real-world model, such as a baghouse that would allow the approximate measurement of the mass flow in extreme conditions, for which there is currently no satisfactory technology on the market. For future research, the authors will perform the validation of the proof of concept shown in this paper, but in a real asphalt plant under real production conditions.

## Figures and Tables

**Figure 1 sensors-23-05019-f001:**
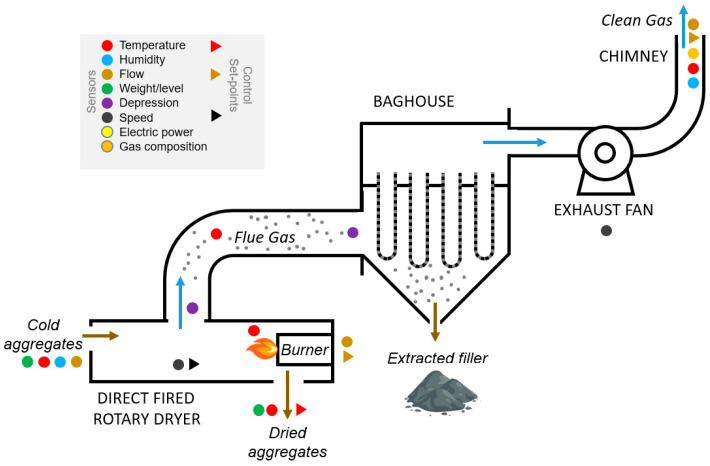
Drying process diagram of an asphalt plant.

**Figure 2 sensors-23-05019-f002:**
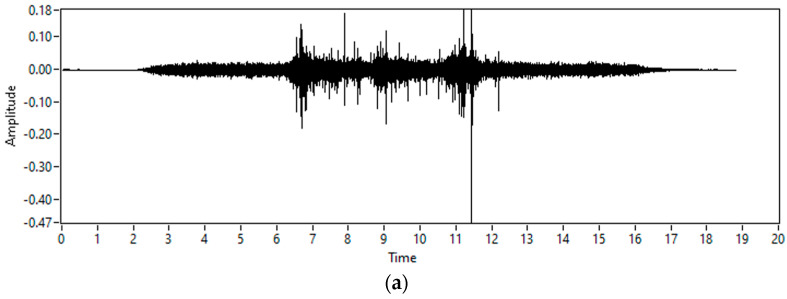
Vibration amplitude values (in g) and spectrogram of three piles of filler.

**Figure 3 sensors-23-05019-f003:**
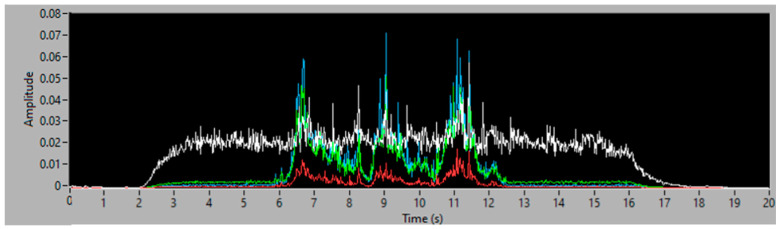
Vibration energy (in g_RMS_) along different frequency bands during filler aspiration.

**Figure 4 sensors-23-05019-f004:**
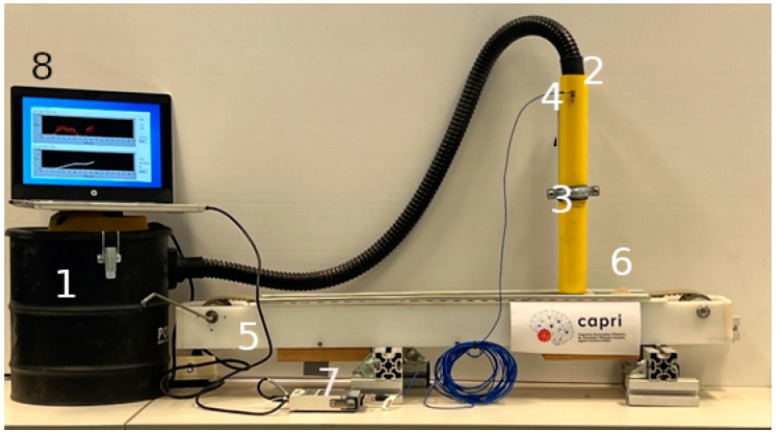
Laboratory simulation of industrial baghouse.

**Figure 5 sensors-23-05019-f005:**
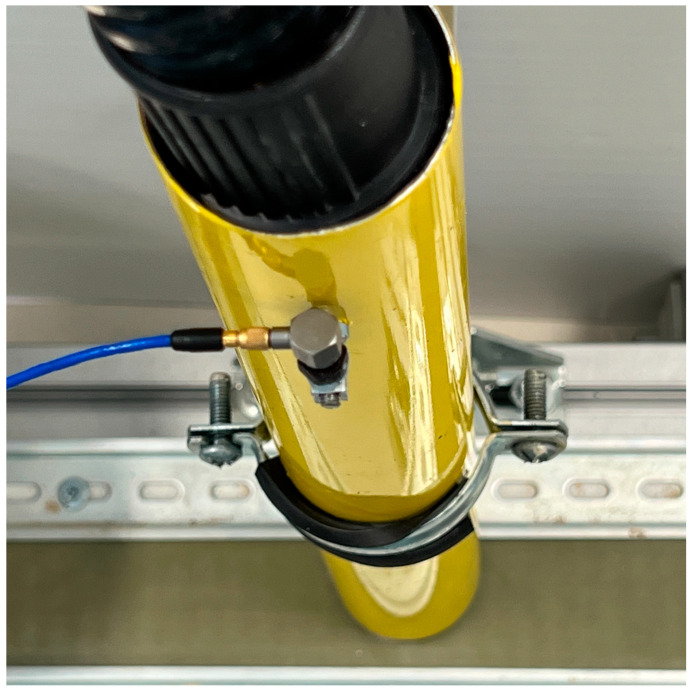
Section of the prototype showing the sensor connected to the slim bar inside the pipe.

**Figure 6 sensors-23-05019-f006:**
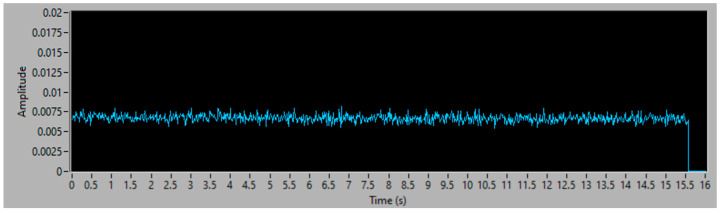
Energy of 14–20 kHz vacuum without filler.

**Figure 7 sensors-23-05019-f007:**
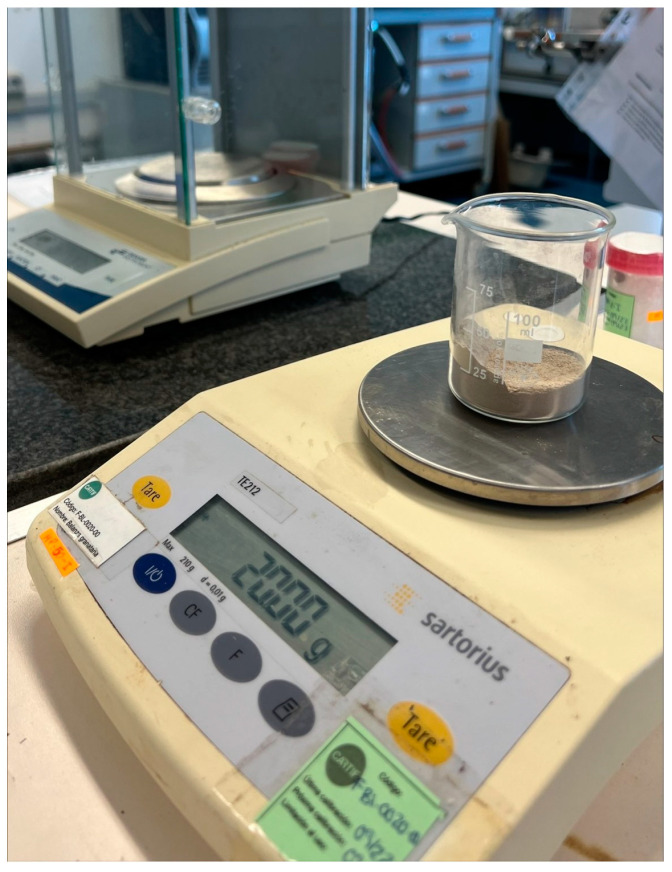
Scale to measure the weight of the filler for the experiments.

**Figure 8 sensors-23-05019-f008:**
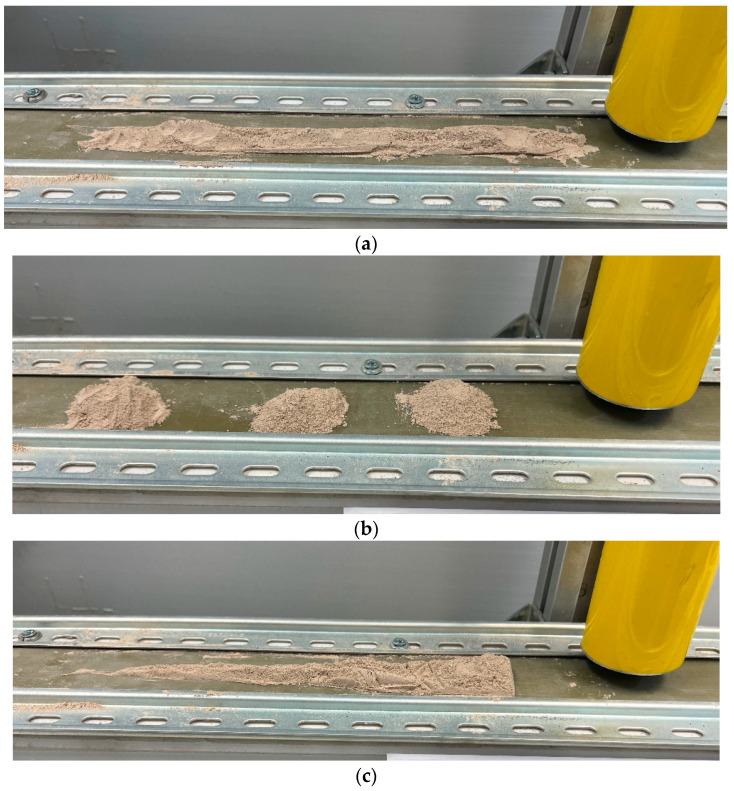
Experiments designed to correlate the total vibration measured and the weight of the filler.

**Figure 9 sensors-23-05019-f009:**
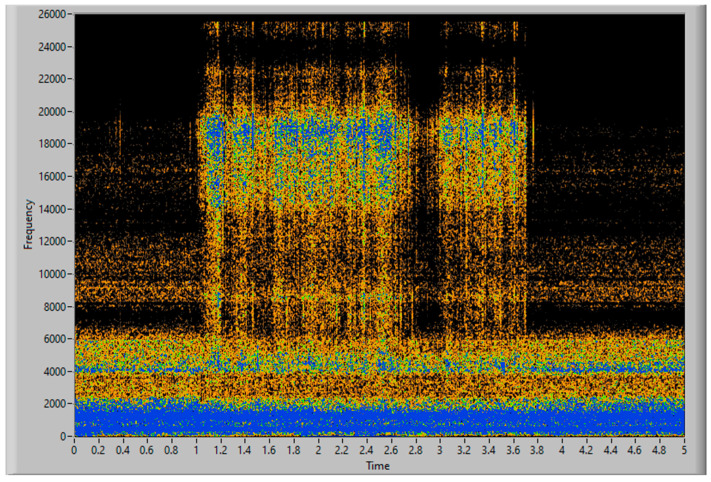
Vibration spectrogram of a continuous strand of 25 g of filler.

**Figure 10 sensors-23-05019-f010:**
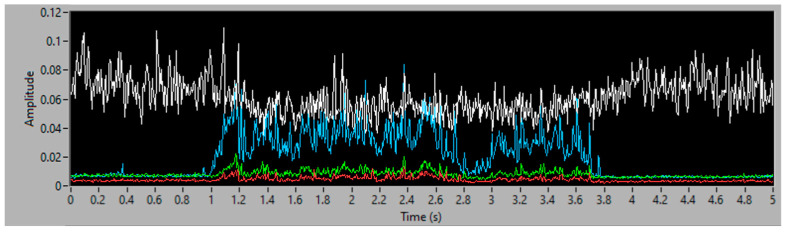
Vibration energy at selected frequency bands of a continuous strand of 25 g of filler.

**Figure 11 sensors-23-05019-f011:**
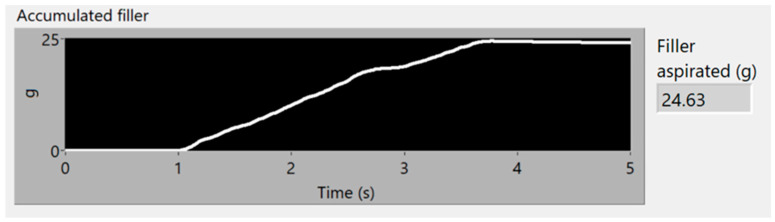
Simulation of grams of aspirated filler.

**Figure 12 sensors-23-05019-f012:**
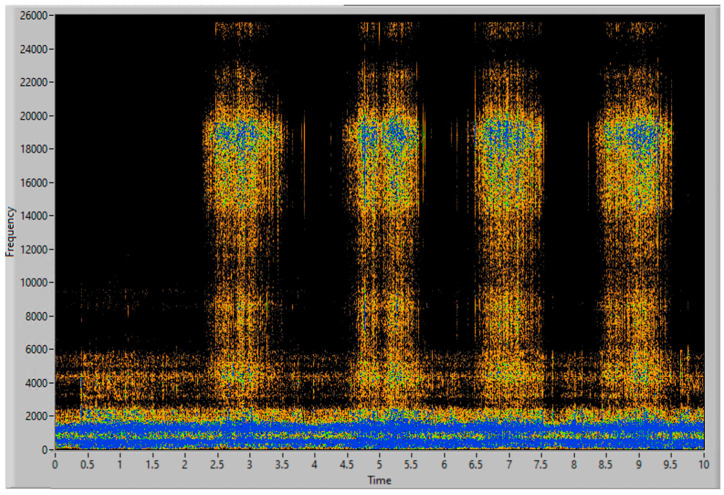
Spectrogram of 60 g of consecutive piles of filler.

**Figure 13 sensors-23-05019-f013:**
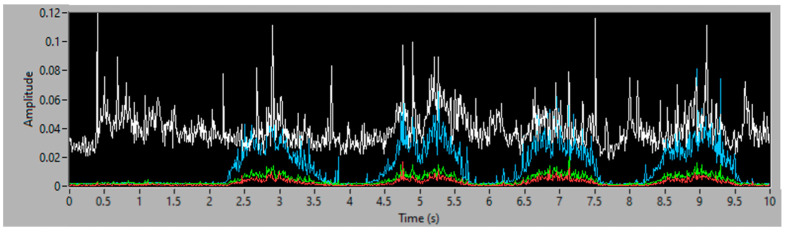
Vibration energy at selected frequency bands of 60 g of consecutive piles of filler.

**Figure 14 sensors-23-05019-f014:**
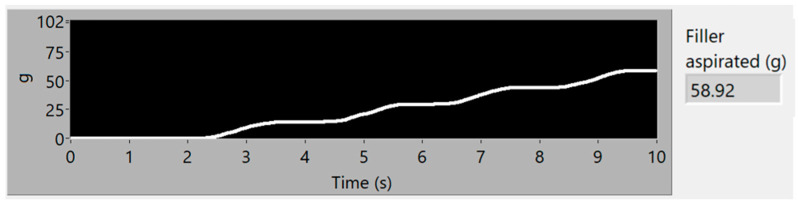
Simulation of the accumulated filler mass, aspirated in 4 piles of 60 g.

**Figure 15 sensors-23-05019-f015:**
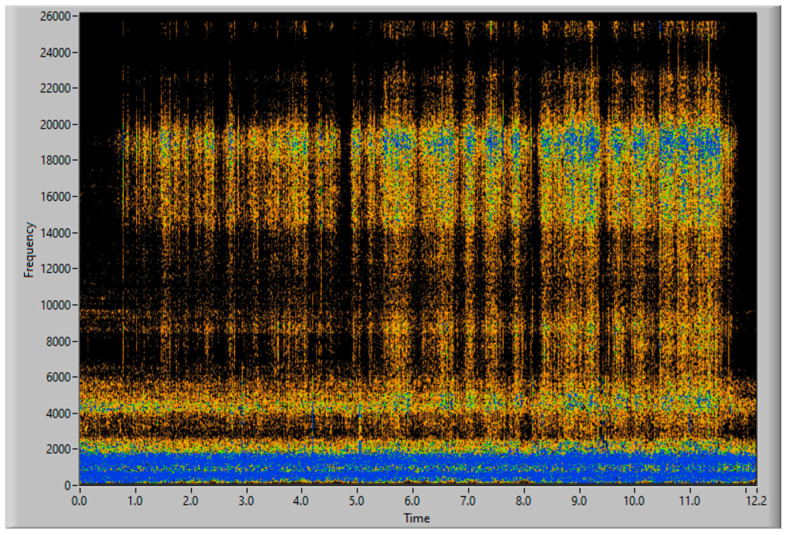
Spectrogram of 100 g of filler, increasingly distributed.

**Figure 16 sensors-23-05019-f016:**
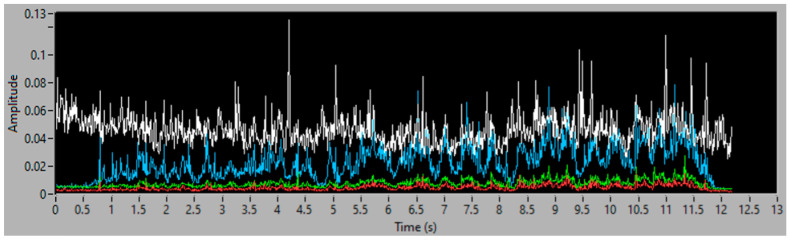
Vibration energy at selected Frequency bands of 100 g of filler, increasingly distributed.

**Figure 17 sensors-23-05019-f017:**
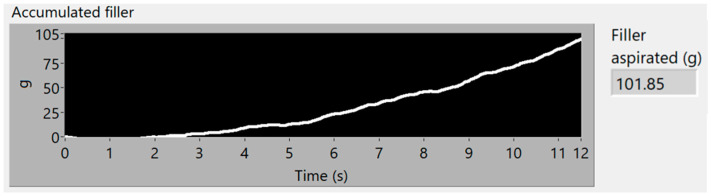
Simulation of the 100 g of aspirated accumulated filler mass, increasingly distributed.

**Figure 18 sensors-23-05019-f018:**
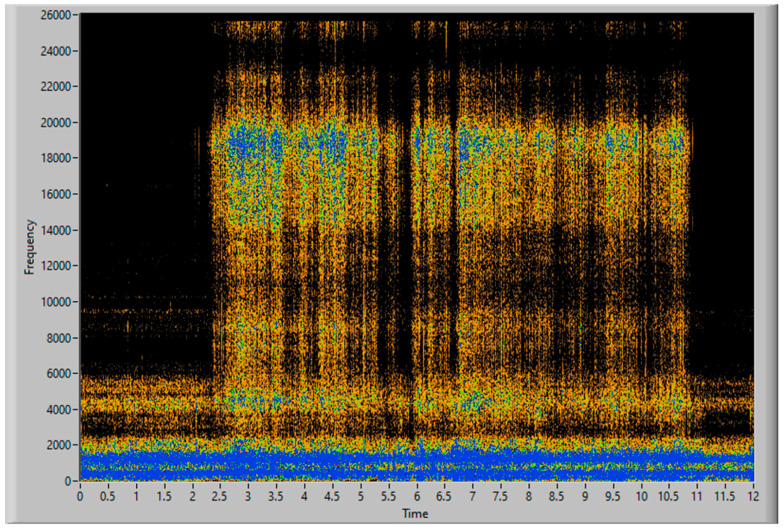
Spectrogram of 100 g of filler, decreasingly distributed.

**Figure 19 sensors-23-05019-f019:**
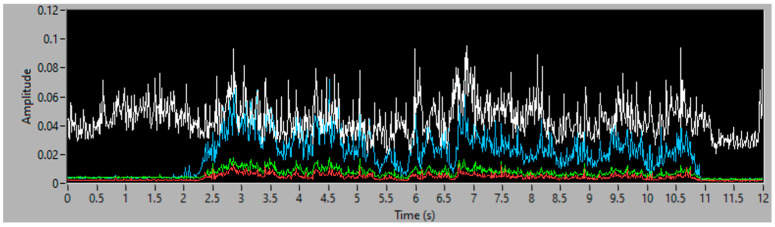
Vibration energy at selected frequency bands of 100 g of filler, decreasingly distributed.

**Figure 20 sensors-23-05019-f020:**
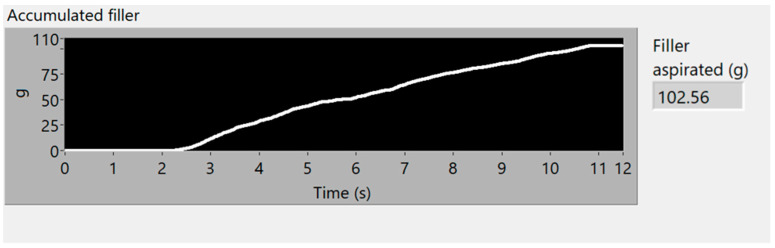
Simulation of the accumulated 100 g of aspirated filler mass, decreasingly distributed.

**Figure 21 sensors-23-05019-f021:**
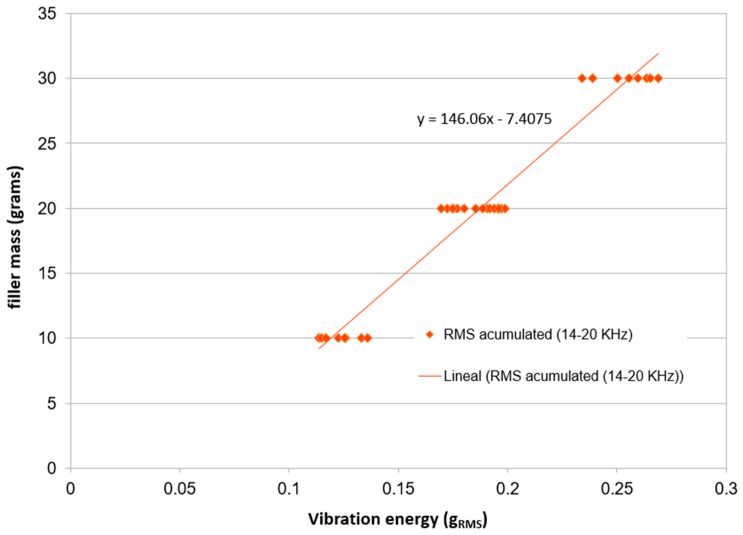
Repeatability and reproducibility of results of aspiration of continuous strands of 10, 20 and 30 g of filler.

**Table 1 sensors-23-05019-t001:** Comparison of actual and estimated values of filler in the four experiments.

Experiment	Actual Weight	Estimated Weight	Error (%)
Continuous strand of filler	25 g	24.63 g	1.48%
Small consecutive piles of filler	60 g	59.82 g	1.80%
Increasing amount of filler	100 g	101.85 g	1.85%
Decreasing amount of filler	100 g	102.56 g	2.56%

## Data Availability

The data of vibration experiments shown in the paper is made available by the authors a the Zenodo repository (https://zenodo.org/record/7787787 (accessed on 21 May 2023)). The dataset contains four files with the raw signal vibration in the HDF5 open data format [18].

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
