# Peer review of "Vibration-Based Smart Sensor for High-Flow Dust Measurement"

_sensors, 2023, doi:10.3390/s23115019_

Round 1

Reviewer 1 Report

The articles describe the Vibration-based smart sensor for high-flow dust measurement. Although the authors have covered all aspects broadly, the following points must be considered for possible publications. 

1. The article needs extensive sentence and grammatical corrections.

2. Line 11: Sentence is incomplete. Change it appropriately.

3. The abstract should not contain references. 

4. The calibration for the proposed sensor is missing.

5. The proposed methodology is not mentioned clearly. Since the authors have mentioned the development of sensors. The fabrication, design, calibration, and evaluation steps are missing. 

Extensive English editing is required. 

Author Response

Reviewer 1

1. The article needs extensive sentence and grammatical corrections. 

The article has been reviewed extensively.

2. Line 11: Sentence is incomplete. Change it appropriately.

The summary has been updated as requested by other reviewer.

3. The abstract should not contain references.

The references have been eliminated from the summary.

4. The calibration for the proposed sensor is missing.

The prototype developed is based on a commercial accelerometer, the explanation of the calibration of the sensor developed to measure the filler, is the transformation from vibration to mass flow  has been enhanced in section 2 about materials and methods.

5. The proposed methodology is not mentioned clearly. Since the authors have mentioned the development of sensors. The fabrication, design, calibration, and evaluation steps are missing.

The section 2 about materials and Methods has been enhanced to show better the methodology of experiments and highlight the measurement principle based on existing commercial sensors (accelerometer). In results section 3 we have created a new subsection to discuss the experiments Results and have added additional experiment results to show replicabiity beyond the 4 types of experiments shown in table 1

Reviewer 2 Report

After having assessed the suitability for publication of the Manuscript titled “Vibration-based smart sensor for high flow dust measurement”, I have distinguished several elements that from my point of view should be made less confusing and more comprehensible by the authors given improving the quality of the manuscript. Therefore, I have devised and written a series of comments to the authors of the manuscript under review.

In this paper, the authors implement a sensor to measure the flow of filler or the finest dust present in the cold aggregates during the drying process. The sensor is based on measuring the generated vibration along a pipe during filler aspiration.

The manuscript is interesting. However, the article under review will be improved if the authors address the following aspects in the text of the manuscript and reflect them point-by-point within the cover letter:

  1. The "Abstract" of the paper. The manuscript will benefit if the authors provide a structured abstract, that covers the following aspects: the background (in which the authors should place the issue that the manuscript addresses in a broad context and highlight the purpose of the study), the methods used to solve the identified issue (that should be briefly described), a summary of the article's main findings followed by the main conclusions or interpretations. In the abstract, the authors must also declare and briefly justify the novelty of their work. The authors must present more clearly the above-mentioned aspects: the background, the methods, the main findings, and the conclusions, as in the actual form of the manuscript, the abstract offers information related only to some of these aspects and even so, their delimitation is unclear. Also, the authors should check the English of the Abstract.
  2. The "Introduction" section – the state of knowledge. In the current form of the manuscript, the authors have performed a survey of what has been done up to this point in the scientific literature. I do not contradict the value of these papers or their relevance in this context, but I consider that the article under review will benefit if the authors explain within the paper what citing criteria are based on which they have chosen the referenced papers.
  3. I recommend the author discuss the repeatability and reproducibility of the measures, by implementing an Anova Gauge R&R experiment.
  4. Discussing the obtained results. When presenting and discussing their obtained results, to validate the contribution and usefulness of the conducted study, the authors should compare their research design, proposed approach, and registered experimental results with the ones obtained by other valuable scientific works.

Author Response

1. The "Abstract" of the paper. The manuscript will benefit if the authors provide a structured abstract, that covers the following aspects: the background (in which the authors should place the issue that the manuscript addresses in a broad context and highlight the purpose of the study), the methods used to solve the identified issue (that should be briefly described), a summary of the article's main findings followed by the main conclusions or interpretations. In the abstract, the authors must also declare and briefly justify the novelty of their work. The authors must present more clearly the above-mentioned aspects: the background, the methods, the main findings, and the conclusions, as in the actual form of the manuscript, the abstract offers information related only to some of these aspects and even so, their delimitation is unclear. Also, the authors should check the English of the Abstract.

We have created a completely new abstract that follows the suggested structure.

2. The "Introduction" section – the state of knowledge. In the current form of the manuscript, the authors have performed a survey of what has been done up to this point in the scientific literature. I do not contradict the value of these papers or their relevance in this context, but I consider that the article under review will benefit if the authors explain within the paper what citing criteria are based on which they have chosen the referenced papers.

We have explained the criteria followed to look for the state of the art for existing commercial solutions, scientific articles and patents highlighting their limitations and which useful features have inspired us to propose the suggested measurement principle.

3. I recommend the author discuss the repeatability and reproducibility of the measures, by implementing an Anova Gauge R&R experiment.

Under new subsection 3.5 about experiments results discussion we have added additional examples of experiments showing the results of several (around 30) repetitions for one type of aspiration use case (continuous line of filler) with different amounts of filler

4. Discussing the obtained results. When presenting and discussing their obtained results, to validate the contribution and usefulness of the conducted study, the authors should compare their research design, proposed approach, and registered experimental results with the ones obtained by other valuable scientific works.

As explained in the state of the art in section 1, the scientific works found in the literature are not directly comparable because they do not apply for the harsh conditions found in the industrial baghouses used in the asphalt manufacturing. 

Round 2

Reviewer 1 Report

Article can be accepted in present form

Reviewer 2 Report

The authors improved the manuscript as needed for publication.